

# Differences in tail feather growth rate in storm-petrels breeding in the Northern and Southern hemisphere: a ptilochronological approach

Anne N.M.A. Ausems, Katarzyna Wojczulanis-Jakubas and Dariusz Jakubas

Department of Vertebrate Ecology and Zoology, Faculty of Biology, University of Gdańsk, Gdańsk, Poland

## ABSTRACT

Moulting and breeding are costly stages in the avian annual cycle and may impose trade-offs in energy allocation between both stages or in their timing. Here, we compared feather growth rates (FGR) of rectrices in adults between two pairs of small pelagic Procellariiformes species differing in moult-breeding strategies: the European storm-petrel *Hydrobates pelagicus* and Leach's storm-petrel *Oceanodroma leucorhoa* breeding in the Northern Hemisphere (Faroe Islands), showing moult-breeding overlap in tail feathers; and the Wilson's storm-petrel *Oceanites oceanicus* and black-bellied storm-petrel *Fregetta tropica*, breeding in the Southern Hemisphere (South Shetlands), temporally separating moult and breeding. We used ptilochronology (i.e., feather growth bar width) to reconstruct FGR reflecting relative energy availability during moult. Based on previous research, we expected positive correlations between feather length (FL) and FGR. Additionally, we expected to find differences in FGR relative to FL between the moult-breeding strategies, where a relatively higher FGR to FL indicates a higher energy availability for moult. To investigate if energy availability during moult in the studied species is similar to species from other avian orders, we used FGR and FL found in literature ($n = 164$) and this study. We fitted a phylogenetic generalized least squares (PGLS) model to FGR with FL, group (i.e., Procellariiformes vs. non-Procellariiformes) and the interaction FL * group as predictors. As it has been suggested that Procellariiformes may form two growth bars per 24 h, we fitted the same model but with doubled FGR for Procellariiformes (PGLSadj). The group term was significant in the PGLS model, but was not in the PGLSadj model, confirming this suggestion. Individually predicted FGR by the PGLSadj model based on FL, showed that the Southern species have a significantly higher FGR relative to FL compared to the Northern species. Additionally, we found no correlation between FL and FGR in the Northern species, and a positive correlation between FL and FGR in the Southern species, suggesting differences in the trade-off between feather growth and size between species from both hemispheres. The observed differences between the Northern and Southern species may be caused by different moult-breeding strategies. The Southern species may have had more energy available for moult as they are free from breeding duties during moult, while the Northern species may have had less free energy due to a trade-off in energy allocation between breeding and moulting. Our study shows how different moult-breeding strategies may affect relative nutritional condition or energy allocation during moult of migratory pelagic seabirds.

Corresponding author
Anne N.M.A. Ausems,
anne.ausems@gmail.com

# INTRODUCTION

Moulting and breeding are energetically costly stages of the annual cycle of birds. The costs of feather synthesis can be illustrated by the fact that metabolic rate during moult increases by more than 100% compared to pre-moulting (*Lindström, Visser & Daan, 1993*). Feather production costs are linked with body mass in a way that moult is relatively more demanding for smaller birds (*Lindström, Visser & Daan, 1993*). Additionally, moult gaps in the remiges and/or rectrices formed after losing old feathers reduce aerodynamic performance, mostly through affecting manoeuvrability (*Hedenström & Sunada, 1999*; *Slagsvold & Dale, 1996*) and less so through increased flight costs (*Hedenström & Sunada, 1999*).

The costs of breeding (e.g., incubation and chick provisioning) are apparent in the increased field metabolic rates (e.g., 11% from incubation to chick rearing in Australasian gannets, *Morus serrator*) (*Green et al., 2013*) and increased stress levels (e.g., higher feather corticosterone concentrations in giant petrels, *Macronectes* spp.) in successful compared to failed breeders (*Crossin et al., 2013*). Increased reproductive costs negatively affected the breeding success in the following year, and birds may even forego breeding if the costs are too high (*Crossin et al., 2013*; *Minguez, 1998*; *Pratte et al., 2018*).

Due to the high energetic costs of moulting and breeding, trade-offs may emerge regarding energy allocation between them, e.g., as shown by decreasing chick quality when artificially increasing parental flight costs (*Mauck & Grubb, 1995*; *Navarro & González-Solís, 2007*). Indeed, in many avian species these two life-stages are temporally separated, with complete moult following the breeding period. Failed breeders and non-breeders often take advantage of the absence of breeding duties by advancing moult (*Alonso et al., 2009*; *Barbraud & Chastel, 1998*; *Crossin et al., 2013*; *Hemborg, Sanz & Lundberg, 2001*; *Mumme, 2018*; *Ramos et al., 2018*). In contrast, individuals that breed relatively late in the season (*Stutchbury et al., 2011*), or that have higher foraging costs during the breeding period (*Alonso et al., 2009*), moult later in the season. Moreover, individuals of some species may suspend moult until they arrive at the wintering areas (*Catry et al., 2013*; *Ramos et al., 2009*), providing some flexibility in allocation of energy between moulting, breeding and migration. The extent of this flexibility partially depends on environmental circumstances (e.g., day-length linked to latitude or food availability) (*Hemborg, Sanz & Lundberg, 2001*; *Terrill, 2018*), and the trade-off between moulting and breeding may even differ strongly between closely related species (e.g., in Northern, *Fulmarus glacialis*, and Southern, *F. glacialoides*, fulmars; *Barbraud & Chastel, 1998*). For instance, some seabird species overlap breeding and moulting, although populations with higher foraging costs show less overlap than populations with lower costs (e.g., in Cory's shearwaters, *Calonectris diomedea borealis*; (*Alonso et al., 2009*). Moult-breeding overlap may therefore only be possible when energetic demands can be met, e.g., when food availability is high (*Alonso et al., 2009*; *Barbraud & Chastel, 1998*). Likewise, moult-breeding overlap seems more

prevalent in sedentary than migratory species (*Bridge, 2006*), though several migratory species adopt this strategy as well (*Alonso et al., 2009*; *Barbraud & Chastel, 1998*; *Ramos et al., 2009*).

Investigating the trade-off in energy allocation between moulting and breeding may be challenging in pelagic seabirds as they are only available for researchers when they come to land for breeding. As at least part of the moulting period is often completed away from the breeding colony, studying their energy management during feather growth may prove difficult. Ptilochronology may offer a way to retrospectively determine the relative amount of energy available during moulting in seabirds, and so evaluate their energy allocation towards feather production. The method is based on feather growth rate, which is determined by the mean feather growth bar width (*Grubb, 1989*; *Grubb, 2006*). Growth bars are alternating light and dark bands formed during feather growth. It is generally assumed that one growth bar is formed over a period of 24 h (*Grubb, 2006*; *Jovani et al., 2011*; *White & Kennedy, 1992*), making it a convenient measure for feather growth rate.

Mean growth bar width is linked with nutritional status, with birds foraging in areas with higher food availability having relatively larger growth bars (*Grubb, 1989*; *Hill & Montgomerie, 1994*). However, within species, growth bar width has also been related to other feather traits (i.e., positively to feather size (*De la Hera, Pérez-Tris & Tellería, 2009*; *Hargitai et al., 2014*; *Le Tortorec et al., 2012*; *Pérez-Tris, Carbonell & Tellería, 2002*), though not in all species (*De la Hera, Pérez-Tris & Tellería, 2009*; *Pérez-Tris, Carbonell & Tellería, 2002*), and negatively to feather quality (*Marzal et al., 2013*). Inter-species comparisons have shown that growth bar width is positively correlated with feather length and mass. This correlation is negatively allometric, such that species with larger feathers have relatively lower growth rates per unit of feather length (*De la Hera, DeSante & Milá, 2012*). A similar correlation has been found between feather growth rate and body size, with larger species having higher absolute feather growth rates, but lower relative growth rates per unit of body size (*Rohwer et al., 2009*).

The aim of our study was to compare relative energy availability during moult between pelagic storm-petrel species with contrasting moult-breeding strategies, i.e., moult-breeding overlap or non-breeding moult. In order to understand the inter- and intra-specific differences in energy availability during moult we compared feather growth rates with feather length. Additionally, to infer the relative energy allocation for each of the species towards moulting, we compared their observed feather growth rate with feather growth rate data for other species found in literature. This study is the first to compare differences in expected feather growth rates between similar species breeding in both hemispheres. Due to their small size and pelagic life-style the non-breeding period of storm-petrels can be hard to study but thanks to recent developments in technology specific migration routes of some species are being discovered (*Pollet et al., 2014*; *Halpin et al., 2018*; *Martínez et al., 2019*; *Lago, Austad & Metzger, 2019*). Our study adds to the understanding of storm-petrel migratory, moulting and breeding strategies by giving some, admittedly indirect, insights into their energy management.

Since larger feathers have been linked to a higher growth rate both within (*De la Hera, Pérez-Tris & Tellería, 2009*; *Hargitai et al., 2014*; *Le Tortorec et al., 2012*; *Pérez-Tris, Carbonell & Tellería, 2002*) and between species (*De la Hera et al., 2011*), we expected to find positive correlations between feather length and growth bar width both within and between the four storm-petrel species. Since the studied species adopt contrasting moult-breeding strategies, we expected to find differences in feather growth rate relative to feather length between the two strategies, indicating differences in relative energy allocation towards moult.

## MATERIALS AND METHODS

### Studied species

We studied European storm-petrels, *Hydrobates pelagicus* (hereafter also ESP), and Leach's storm-petrels, *Oceanodroma leucorhoa* (hereafter also LSP), breeding sympatrically in the Northern Atlantic, and Wilson's storm-petrels, *Oceanites oceanicus* (hereafter also WSP), and black-bellied storm-petrels, *Fregetta tropica* (hereafter also BBSP), breeding sympatrically in the Maritime Antarctic. The European storm-petrel is the world's smallest pelagic seabird, while the Wilson's storm-petrel is the smallest endotherm breeding in the Antarctic. Black-bellied and Leach's storm-petrels are similar in body morphology, apart from tarsus length, and both are significantly larger than the European and Wilson's storm-petrels (*Carboneras et al., 2017*). All four species are migratory, and move towards and sometimes beyond the equator, during the non-breeding season. Though morphologically similar (*Flood & Thomas, 2007*), storm-petrels are divided into two families: the Northern *Hydrobatidae* and the Southern *Oceanitidae* (*Penhallurick & Wink, 2004*; *Rheindt & Austin, 2005*; *Robertson et al., 2016*).

The breeding season for all species takes several months from first arrival at the colony to fledging and takes place during summer (boreal and austral in Northern and Southern hemispheres respectively), with chicks fledging in late summer (egg laying until fledging takes on average 3,5 months for all species) (*Carboneras et al., 2017*; *Cramp et al., 1977*; *Wasilewski, 1986*). The diets of the studied storm-petrel species consist mostly of crustaceans and myctophid fish, though the Northern species eat relatively more fish than crustaceans compared to the Southern species (*Ainley, O'Connor & Boekelheide, 1984*; *Ainslie & Atkinson, 1936*; *Büßer, Kahles & Quillfeldt, 2004*; *Croxall & North, 1988*; *Croxall & Prince, 1980*; *D'Elbée & Hémery, 1998*; *Hahn et al., 1998*; *Hedd & Montevecchi, 2006*; *Quillfeldt, 2002*; *Ridoux, 1994*; *Wasilewski, 1986*). Wilson's and black-bellied storm-petrels start moulting after the breeding period (*Beck & Brown, 1972*) while European and Leach's storm-petrels start moulting during the breeding period, exhibiting moult-breeding overlap (*Ainley, Lewis & Morrell, 1976*; *Amengual et al., 1999*; *Arroyo et al., 2004*; *Bolton & Thomas, 2001*).

### Sample collection

We sampled European ($n = 52$) and Leach's storm-petrels ($n = 55$) in the Northern Hemisphere (hereafter Northern species) on the island of Mykines, Faroe Islands (62°05′N, 07°39′W). During the breeding period of 2018 we captured adults in mist nets at night,
placed in a mixed colony. We studied Wilson's ($n = 228$) and black-bellied storm-petrels ($n = 32$) in the Southern Hemisphere (hereafter Southern species), on King George Island, South Shetland Islands, Antarctica (62°09′S, 58°27′W). During the breeding periods of 2017 and 2018 we captured adults in mist nets placed in the colonies and took parents from the nests.

We collected the right outermost rectrix from adults of the four species of storm-petrels. In 2018 32 adults were recaptured that were previously caught in 2017, with fully formed rectrices. Additionally, one Wilson's storm-petrel was recaptured within 2018 with a fully regrown rectrix, though the regrown feather has not been used for the statistical analyses. We did not notice anything untoward in their tail feathers, or during the analyses (e.g., obvious outliers), which leads us to assume that our plucking of the feathers did not cause long-term harm to the birds. See below for pseudo-replication management.

All individuals of the Northern species and some individuals of the Southern species were captured in mist-nets, which could lead to uncertainty in the breeding stage of the adults. By capturing birds in a mist-net it becomes harder to determine the breeding status of the sampled adults, as sub-adults may be caught while prospecting the colony (floaters) (*Sanz-Aguilar et al., 2010*). Especially when using tape-lures, prospecting birds may be attracted to the net (*Furness & Baillie, 1981*; *Amengual et al., 1999*). However, breeding birds can be identified by their readiness to regurgitate and the presence of a brood patch (*Furness & Baillie, 1981*). We did not use tape-lures for the European storm-petrels, or either of the Southern species, during capturing events, decreasing the likelihood of catching floaters. We did use tape-lures for the Leach's storm-petrels, which could have increased the chances of attracting floaters. However, almost all Leach's storm-petrels were observed to readily regurgitate, and all had either fully bare brood patches, or brood patches with only few feathers present. This leads us to assume that at least the vast majority of the sampled birds were breeders.

Birds were handled under licence of the Statens Naturhistoriske Museum, Københavns Universitet C 1012 and with permission of the Polish National SCAR, Institute of Biochemistry and Biophysics (Permit for entering the Antarctic Specially Protected Area No. 3/2016 & No. 08/2017, Permit for taking or harmful interference of Antarctic fauna and flora No. 6/2017 & No. 7/2016). Permission to enter the study site on Mykines was sought through local land-owners.

### Feather measurements

We measured feather length (FL) from the tip to the base of the calamus with calipers to the nearest 0.1 mm. We measured growth bar width by placing the feather on a white paper background and marking the tip and the base of the calamus, and each visible growth bar in the vane area of the feather before rounding of the tip and above the white area, with a pinprick. We then used calipers to measure the distances between each pinprick on the background to the nearest 0.1 mm, following *Grubb (1989)*. A new piece of paper was used for each feather. We used mean growth bar width per feather as a proxy for feather growth rate (FGR).

## Statistical analyses

Since we sampled the Southern species during two field seasons, we investigated the inter-annual differences in FGR and FL using a Welch $t$-test ($t.test$, package $stats$ in R version 3.6.1 ($R$ $Core$ $Team$, $2018$)). FGR was significantly higher in 2017 compared to 2018 for the Wilson's but did not differ significantly for black-bellied storm-petrels (Welch $t$-test; WSP: $t_{163.69} = 3.192$, $p = 0.002$; BBSP: $t_{29.343} = -0.901$, $p = 0.375$). However, although significant for the Wilson's storm-petrels, we deemed the absolute differences in FGR between the years small enough (high overlap of the 95% confidence ellipses, Fig. S1) to justify pooling the data. FL did not differ significantly between the years for either species (Welch $t$-test; WSP: $t_{216.29} = -0.549$, $p = 0.584$; BBSP: $t_{29.706} = -0.519$, $p = 0.608$), and therefore we also pooled these data.

Since some individuals were caught in both years ($n = 28$ for WSP, $n = 3$ for BBSP) we assessed the effect of pseudo-replication by comparing the mean values of FL and FGR between the seasons of 2017 and 2018 for the Wilson's and black-bellied storm-petrels individuals captured in both years. We found no significant differences between the means of both seasons for either species (Paired $t$-test; FL: WSP: $t_{27} = -0.993$, $p = 0.330$; BBSP: $t_2 = 0.096$, $p = 0.932$; FGR: WSP: $t_{27} = 1.469$, $p = 0.153$; BBSP: $t_2 = -1.023$, $p = 0.414$). Thus, in further analyses based on individuals, to avoid pseudo-replication, we used the mean values per individual instead of repeated measurements which reduced the sample size to $n = 200$ unique individuals for the Wilson's storm-petrels and to $n = 29$ unique individuals for the black-bellied storm-petrels.

To compare FL and FGR among species we used univariate tests. Due to inequality of variances (Fligner-Killeen test, $fligner.test$, package $stats$) of FL ($\chi^2 = 10.87$, df=3, $p = 0.012$) we used non-parametric Kruskal–Wallis and $post$-$hoc$ Dunn tests ($dunn.test$, package $dunn.test$) for all inter-species comparisons. To examine the relationships between FL and FGR for each species, we used Spearman's rho correlation ($cor.test$, package $stats$) because we did not necessarily expect linear relationships after plotting FGR and FL data for multiple species found in literature (Fig. S2). Additionally, we chose not to transform the data to make them linear as the transformations needed differed between the species and would inhibit inter-specific comparisons.

To investigate if the observed FGR of the studied species was higher or lower than expected (i.e., what their energy availability was) we fitted a phylogenetic generalized least square (PGLS) model ($gls$, package $nlme$) with Pagel's λ ($corPagel$, package $ape$) to multi-species data. The full model contained FGR as response variable with FL, group [group 1: non-Procellariiformes ($n = 162$); group 2: Procellariiformes ($n = 6$)] and the interaction FL * group as predictor variables (PGLS model). We used ∆AIC to determine if the updated model had a better fit, and dropped terms that did not improve the model.

For the PGLS model we used FGR and FL data found in literature ($n = 164$ species, 194 observations) and from this study ($n = 4$ species) (Table S1). For species with multiple records of FGR and FL, we averaged the values per species. We searched for suitable studies in the Web of Science Database (https://www.webofknowledge.com; 05-11-2018) using $ptilochronology$, $growth$ $bars$ and $feather$ $growth$ $rate$ as keywords. We then only selected papers if they contained FGR and FL measurements in SI units.

We reconstructed the phylogeny based on the most recent complete avian time-calibrated phylogeny (*Jetz et al., 2012*) with a backbone tree developed by *Ericson et al. (2006)*. To account for phylogenetic uncertainty we calculated the consensus tree, based on 100 alternative trees, downloaded from the BirdTree database (http://www.birdtree.org; *Jetz et al., 2012*). We corrected for FGR and FL left-skewed data by log10 transformation of the data.

The feather types (rectrix or primary) used to determine FGR differed between studies, but FGR is highly correlated between both types (*Saino et al., 2012*). A comparison of correlation coefficients for a PGLS model with only rectrices ($n = 129$) and a PGLS model with only primaries ($n = 44$), using Fisher's Z (*cocor.indep.groups*, package *cocor*) showed no significant difference ($z = 1.91$, $p = 0.056$) (Table S2). Feather type was thus not used as a predictor in the PGLS models.

*Langston & Rohwer (1996)* suggested that in the Procellariiformes the relationship between FGR and FL may differ from that of other species, i.e., they may form two growth bars per 24 h due to foraging on prey that show diel migration. To test this possibility we firstly ran the PGLS model with raw data. Then, we fitted identical PGLS models (i.e., FGR $\sim$ FL, FGR $\sim$ FL + Group and FGR $\sim$ FL * Group) but doubled the FGR values for the Procellariiformes (PGLSadj), and again dropped terms that did not improve the model. We could not compare the models with the raw and adjusted data directly with each other, as they have different data sets, but with this approach we could show the effect of Group on the model fit for both data sets.

We predicted FGR based on individual FL inserted into the PGLSadj model, and then calculated the residual difference with observed FGR doubled. As Fligner-Killeen tests showed variance inequality in the residuals between the species ($\chi^2 = 24.339$, $df = 3$, $p < 0.001$) and hemispheres ($\chi^2 = 26.077$, $df = 3$, $p < 0.001$) we used non-parametric Kruskal–Wallis and Dunn post-hoc tests to compare the differences between the species. To compare the residuals between the two hemispheres, and thus moult-breeding strategies, we used Welch two-sample $t$-tests as they are robust for variance differences (*t.test*, package *stats*). To determine whether the residuals where positively or negatively different from zero, and thus if energy availability was relatively high or low, we used one-sample Student's $t$-tests for each species (*t.test*, package *stats*).

## RESULTS

### Feather characteristics

We found significant differences between the species in FGR and FL (Kruskal–Wallis test, FGR: $\chi^2 = 214.35$, $p < 0.001$; FL: $\chi^2 = 248.35$, $p < 0.001$). Post-hoc tests (Dunn test, $p < 0.001$) revealed that black-bellied storm-petrels had a higher FGR than Wilson's storm-petrels and the Southern species had a higher FGR than the Northern species (Table 1). FL differed significantly between all species pairs (Dunn-test, $p < 0.001$) except between black-bellied and Leach's storm-petrels (Table 1). Only the Southern species showed a significant positive correlation between FGR and FL, with Wilson's storm-petrels showing a weak positive correlation (Spearman correlation, $r_s = 0.215$, $p = 0.002$) and

**Table 1   Results of the *post-hoc* Dunn test for inter-specific differences in feather length (FL) and feather growth rate (FGR) for each studied species.**

| Species | Var | Black-bellied storm-petrel (Z, *p*) | | European storm-petrel (Z, *p*) | | Leach's storm-petrel (Z, *p*) | |
|---|---|---|---|---|---|---|---|
| European storm-petrel | FL | 12.177, | <0.001 | | | | |
| | FGR | 10.363, | <0.001 | | | | |
| Leach's storm-petrel | FL | 0.698, | 0.243 | −13.763, | <0.001 | | |
| | FGR | 9.438, | <0.001 | −1.218, | 0.111 | | |
| Wilson's storm-petrel | FL | 7.587, | <0.001 | −8.446, | <0.001 | 8.850, | <0.001 |
| | FGR | 3.602, | <0.001 | −10.831, | <0.001 | −9.528, | <0.001 |

Notes.

Var, variable, *p*-values $\leq \alpha/2(\alpha = 0.05)$ are bolded.

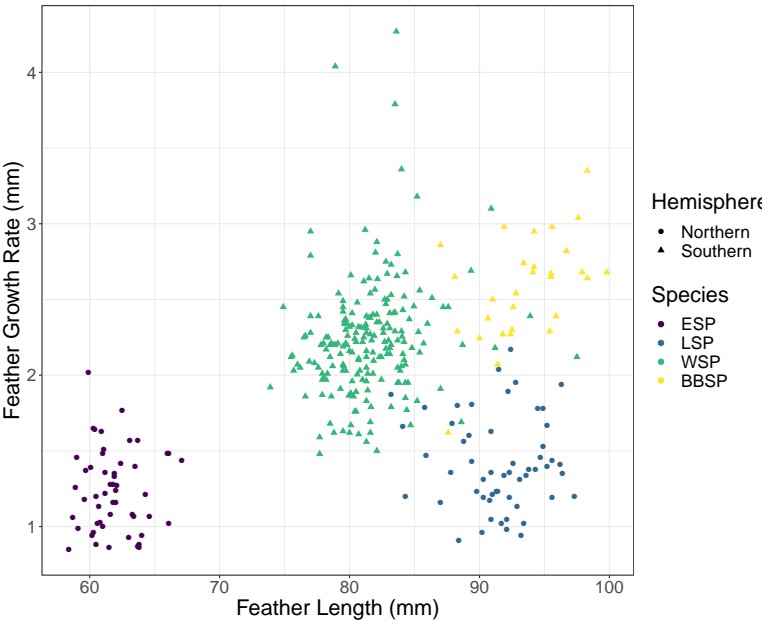

**Figure 1   Correlation between feather growth rate (FGR) and feather length (FL) for all four studied storm-petrel species.** European storm-petrels (ESP) are shown in purple; Leach's storm-petrels (LSP) in blue; Wilson's storm-petrels (WSP) in green; black-bellied storm-petrels (BBSP) in yellow. Species from the Northern Hemisphere are shown with dots, species from the Southern Hemisphere with triangles. See Tables 1 and 2 for statistical analyses.

black-bellied storm-petrels a moderately positive correlation ($r_s = 0.513$, $p = 0.04$) (Fig. 1, Table 2). For the Northern species we found no correlation between FGR and FL ($p > 0.05$) (Table 2).

### Relative energy availability

In the full PGLS model (AIC = −497.59), FL ($p < 0.001$) and group (i.e., Procellariiformes vs. non-Procellariiformes) had a significant effect ($p < 0.001$) on FGR but the interaction FL * group did not ($p = 0.729$) (Fig. 2A, Table 3). The PGLS model with group as a predictor (hereafter optimised PGLS model) was better (AIC = −499.47) than the PGLS

**Table 2 Spearmans Rank Correlation output for correlations between feather growth rate (FGR) and feather length (FL) for each studied species.**

| Species | S | rho | p-value |
|---|---|---|---|
| European storm-petrel | 2,3327 | 0.004 | 0.976 |
| Leach's storm-petrel | 27,756 | −0.001 | 0.993 |
| Wilson's storm-petrel | 1,046,299 | 0.215 | **0.002** |
| black-bellied storm-petrel | 1,977.7 | 0.513 | **0.004** |

**Notes.**

S, sum of squared rank differences; rho, Spearmans rank correlation rho.

$P$-values $\leq 0.05$ are bolded.

model without group (AIC $= -484.99$, $\Delta$AIC $= 14.48$) (Table 4). The model with group and interaction did not differ from the model without the interaction (AIC $= -497.59$, $\Delta$AIC $= 1.88$). After multiplying Procellariiformes' FGR by two (i.e., PGLSadj), FL still had a significant effect on FGR ($p < 0.001$) (Fig. 2B, Table 3). Neither group ($p = 0.912$) nor the interaction FL $^{\star}$ group was significant ($p = 0.729$) (Table 3). The PGSLadj model without the group predictor (AIC $= -501.46$) was not different from the PGLSadj model including group (AIC $= -499.47$, $\Delta$AIC $= 1.99$) nor was the PGLSadj model including group different from the model including group and interaction (AIC $= 497.59$, $\Delta$AIC $= 1.88$) (Fig. 2B, Table 4).

The individual residuals of predicted FGR based on the optimized PGLS model differed significantly between the studied storm-petrel species (Kruskal–Wallis; $\chi^2 = 198.92$, $df = 3$, $p < 0.001$)(Table 5) and hemispheres (Welch $t$-test, $p < 0.001$)(Fig. 3). The residuals for the Southern species were significantly higher than those of the Northern species (mean Southern $= 0.126$, mean Northern $= -0.091$, $t_{158.56} = -21.371$). The residuals differed significantly from zero for all four species. Both Northern species had negative residuals, while both Southern species had positive residuals (Student's $t$-test, ESP: $t_{51} = -5.847$, $p < 0.001$; LSP: $t_{54} = -8.367$, $p < 0.001$; WSP: $t_{199} = 25.310$, $p < 0.001$; BBSP $t_{28} = 14.460$, $p < 0.001$) (Fig. 3, Table 6).

# DISCUSSION

The Spearman's rho correlations showed significant, positive relationships between mean growth bar width and feather length for the Southern storm-petrel species, but not for the Northern species. Additionally, using the PGLS model, we found that the Southern species had a higher feather growth rate than predicted while the Northern species had a lower feather growth rate than predicted.

The difference in residual length between the studied species, and between the hemispheres may be associated with a difference in relative energy availability during moulting between species of both hemispheres, possibly caused by their different moult-breeding strategy. The Southern species, both moulting during the non-breeding period (*Beck & Brown, 1972*), are free from breeding duties during moult and may use all available energy for feather synthesis, while the Northern species, showing moult-breeding overlap (*Amengual et al., 1999*; *Arroyo et al., 2004*; *Bolton & Thomas, 2001*), have to allocate that energy between moulting and breeding. The differences between storm-petrels from both

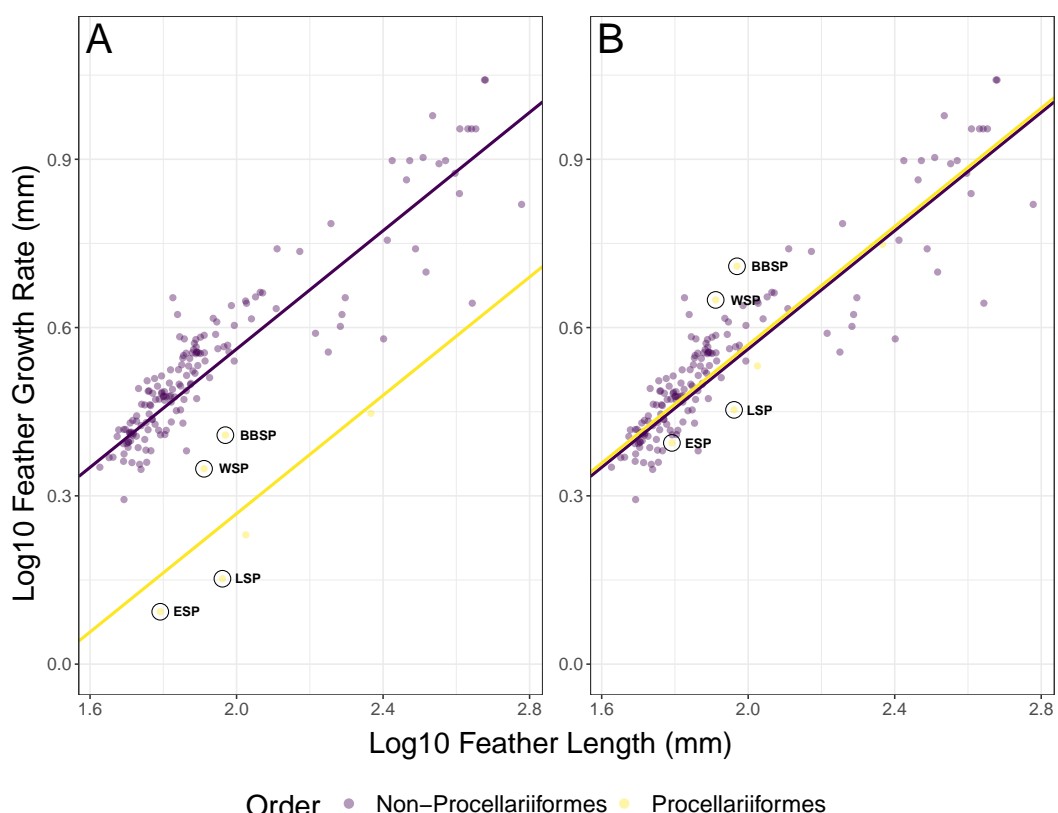

**Figure 2  Phylogenetic generalized least squares (PGLS) models for feather growth rate (FGR) based on feather length (FL).** (A) The optimized PGLS model, with Pagel's λ , based on the phylogenetic tree, was fitted to log10 FGR as response variable, and log10 FL and group as predictors, for data found in literature ($n = 164$ species) and this study ($n = 4$ species). The groups were defined as non-Procellariiformes and Procellariiformes, to determine whether the Procellariiformes behaved differently from the other reported species. The data used for this model were not adjusted. (B) The PGLS model with group as predictor and with Procellariiformes FGR doubled (PGLSadj model), following *Langston & Rohwer*'s (*1996*) suggestion that Procellariiformes might form two GBs per 24 h. The group term was not significant in this model, indicating that the aforementioned suggestion was likely correct. Procellariiformes are shown in yellow, non-Procellariiformes in purple. The studied species are circled in black (ESP, European storm-petrel; LSP, Leach's storm-petrel; WSP, Wilson's storm-petrel; BBSP, black-bellied storm-petrel). See also Table 3 for model description and Table 4 for model comparison.

hemispheres in trade-offs in energy allocation between moulting and breeding may affect the correlations between feather length and feather growth rate, resulting in a lack of significant correlations between feather length and mean growth bar width in the Northern species in contrast to significant relationships between feather length and mean growth bar width in the Southern species.

Moult-breeding overlap in the Northern species has so far only been shown in the Mediterranean subspecies of the European storm-petrel (*Hydrobates pelagicus melitensis*) (*Amengual et al., 1999*; *Arroyo et al., 2004*), in a British population of European storm-petrels (Scott 1970 in *Cramp et al., 1977*), in Canadian populations of the Leach's storm-petrel (*Ainley, Lewis & Morrell, 1976*), but the overlap extend is not (yet) generally accepted

**Table 3  Phylogenetic generalized least squares (PGLS) models for feather growth rate (FGR) based on feather length (FL).** PGLS models, with Pagel's λ based on the phylogenetic tree, were fitted to log10 FGR as response variable and log10 FL as predictor, for data found in literature ($n = 164$ species) and this study ($n = 4$ species). To determine whether the Procellariiform species considered ($n = 6$ species) differed in number of growth bars (GB) formed per 24 h, we added a group term (group 1 = non-Procellariiformes, group 2 = Procellariiformes) and its interaction to the full PGLS model (no. 1). Terms were dropped based on significance and improvement of AIC (no. 2 & 3). To test whether Langston and Rohwer's (1996) suggestion that Procellariiformes might form two GBs per 24 h was true, Procellariiformes FGR was doubled (PGLSadj model) and an analogous set of models were tried. Pagel's λ is the phylogenetic signal, with values between 0 and 1.

| Model | No. | Predictor | AIC | Pagel's λ | Estimate | SE | *t*-value | Pr(>|t|) |
|---|---|---|---|---|---|---|---|---|
| PGLS | 1 | Intercept | −497.59 | 0.935 | −0.500 | 0.107 | −4.669 | **<0.001** |
| | | Log10(FL) | | | 0.531 | 0.043 | 12.320 | **<0.001** |
| | | Group | | | −0.156 | 0.402 | −0.388 | 0.699 |
| | | Log10(FL):Group | | | −0.067 | 0.194 | −0.347 | 0.729 |
| | 2 | Intercept | −499.47 | 0.935 | −0.493 | 0.105 | −4.708 | **<0.001** |
| | | Log10(FL) | | | 0.527 | 0.042 | 12.588 | **<0.001** |
| | | Group | | | −0.294 | 0.068 | −4.309 | **<0.001** |
| | 3 | Intercept | −484.99 | 0.962 | −0.590 | 0.112 | −5.288 | **<0.001** |
| | | Log10(FL) | | | 0.554 | 0.045 | 12.374 | **<0.001** |
| PGLSadj | 1 | Intercept | −497.59 | 0.935 | −0.500 | 0.107 | −4.669 | **<0.001** |
| | | Log10(FL) | | | 0.531 | 0.043 | 12.320 | **<0.001** |
| | | Group | | | 0.145 | 0.402 | 0.361 | 0.719 |
| | | Log10(FL):Group | | | −0.067 | 0.194 | 0.347 | 0.729 |
| | 2 | Intercept | −499.47 | 0.935 | −0.493 | 0.105 | −4.708 | **<0.001** |
| | | Log10(FL) | | | 0.527 | 0.042 | 12.588 | **<0.001** |
| | | Group | | | 0.008 | 0.068 | 0.110 | 0.912 |
| | 3 | Intercept | −501.46 | 0.935 | −0.490 | 0.102 | −4.814 | **<0.001** |
| | | Log10(FL) | | | 0.527 | 0.041 | 12.760 | **<0.001** |

**Notes.**
All *p*-values ≤ 0.05 are bolded.

**Table 4  AIC and ΔAIC values of phylogenetic generalized least squares (PGLS) models.** A PGLS model with Pagel's λ was fitted to log10 feather growth rate (FGR) with feather length (FL) and Group (non-Procellariiformes vs. Procellariiformes). No 1 is the full model, 2 & 3 with the interaction, and the interaction and group dropped respectively. Models were fitted to raw data (PGLS model) and data with Procellariiformes feather growth rate doubled (PGLSadj). Model selection was based on AIC and delta, but properly as I can't seem to add it to acrobat reader AIC values.

| Model | No. | df | AIC | ΔAIC |
|---|---|---|---|---|
| PGLS | 1 | 6 | −497.59 | 1.88 |
| | 2 | 5 | −499.47 | 0.00 |
| | 3 | 4 | −484.99 | 14.48 |
| PGLSadj | 1 | 6 | −497.59 | 1.88 |
| | 2 | 5 | −499.47 | 1.99 |
| | 3 | 4 | −501.46 | 0.00 |

in Northern populations. However, preliminary stable-isotope analyses show that tail feather isotopes of the Northern species are more closely matched with blood isotopes collected during the breeding season, than those of the Southern species (Ausems et al., in prep.). This seems to indicate that both the feathers and blood were synthesised under more

**Table 5  Results of non-parametric post-hoc Dunn test for the residuals of the storm-petrel species from the optimized pylogenetic general-ized least squares (PGLS) model.** For the multi-species model, PGLS models were fitted to log10 feather growth rate (FGR) found in literature as response variable with log10 feather length (FL) as predictor. To test wether the Procellariiformes behaved differently from rother species, a group variable was added with group 1 being all non-Procellariiformes and group 2 being the Procellariiformes. The residuals of the predicted log10 FGR of the individuals per storm-petrel species were obtained by inserting individual FL into the model with group set to Procellariiformes, and compar-ing the predicted log10 FGR with log10 observed FGR. $P$-values $\leq \alpha/2(\alpha = 0.05)$ are bolded. For plots see Fig. 3.

| | Dunn's multiple comparisons test | | | | | |
|---|---|---|---|---|---|---|
| Species | black-bellied storm-petrel (Z, p) | | European storm-petrel (Z, p) | | Leach's storm-petrel (Z, p) | |
| European storm-petrel | 8.389, | **<0.001** | | | | |
| Leach's storm-petrel | 9.075, | **<0.001** | 0.715, | 0.237 | | |
| Wilson's storm-petrel | 2.228, | **0.013** | −9.646, | **<0.001** | −10.770, | **<0.001** |

**Table 6  One-sample Student's $t$-test output of the residuals of feather growth rate (FGR) of each studied species predicted by the optimized phylogenetic generalized least squares (PGLS) model.** For the multi-species model, PGLS models were fitted to log10 feather growth rate (FGR) found in literature as response variable with log10 feather length (FL) as predictor. To test wether the Procellariiformes behaved differently from re-sults reported for other species, a group variable was added with group 1 being all non-Procellariiformes and group 2 being the Procellariiformes. The residuals of the predicted log10 FGR of the individuals per storm-petrel species were obtained by inserting individual FL into the model with group set to Procellariiformes, and comparing the predicted log10 FGR with log10 observed FGR. $P$-values $\leq \alpha/2(\alpha = 0.05)$ are bolded. For plots see Fig. 3.

| Species | Residuals | | | Residuals difference | | |
|---|---|---|---|---|---|---|
| | Mean residuals | 95% CI lower | 95% CI upper | t | df | p |
| European storm-petrel | −0.076 | −0.102 | −0.050 | −5.847 | 51 | **<0.001** |
| Leach's storm-petrel | −0.106 | −0.132 | −0.081 | −8.367 | 54 | **<0.001** |
| Wilson's storm-petrel | 0.122 | 0.113 | 0.132 | 25.310 | 199 | **<0.001** |
| black-bellied storm-petrel | 0.154 | 0.132 | 0.176 | 14.460 | 28 | **<0.001** |

similar foraging conditions and in similar foraging areas, strengthening our conviction that the Northern species at least partially overlap their tail moult and breeding.

Austral summer is short and primary production is highest only in favourable conditions (i.e., longer daylight hours and retreating sea ice) (*Arrigo, Van Dijken & Bushinsky, 2008*; *Murphy et al., 2016*). The peak abundance of the main prey of the Southern storm-petrels, Antarctic krill, *Euphausia superba*, usually lasts from December to February (*Food and Agriculture Organization of the United Nations, 2019*; *Ross & Quetin, 2014*). The relatively short period of high food abundance and possible competition over it e.g., from penguins, whales and krill fisheries (*Barlow et al., 2002*; *Descamps et al., 2016*; *Ratcliffe et al., 2015*), could inhibit the Southern storm-petrels from overlapping moult and breeding as there is no longer enough food available at the end of the breeding season. Additionally, compared to the North Atlantic and Arctic ocean, the highly productive oceanographic features in the Southern Ocean, such as ocean fronts and eddies, occur over larger spatial scales and are usually farther away from the breeding colonies, forcing the birds to take longer foraging trips (*Bost et al., 2009*). During the non-breeding period birds free from the constraint of central-place foraging may exploit these highly productive areas freely, which may explain the relatively higher than predicted daily feather growth rate of the Southern species.

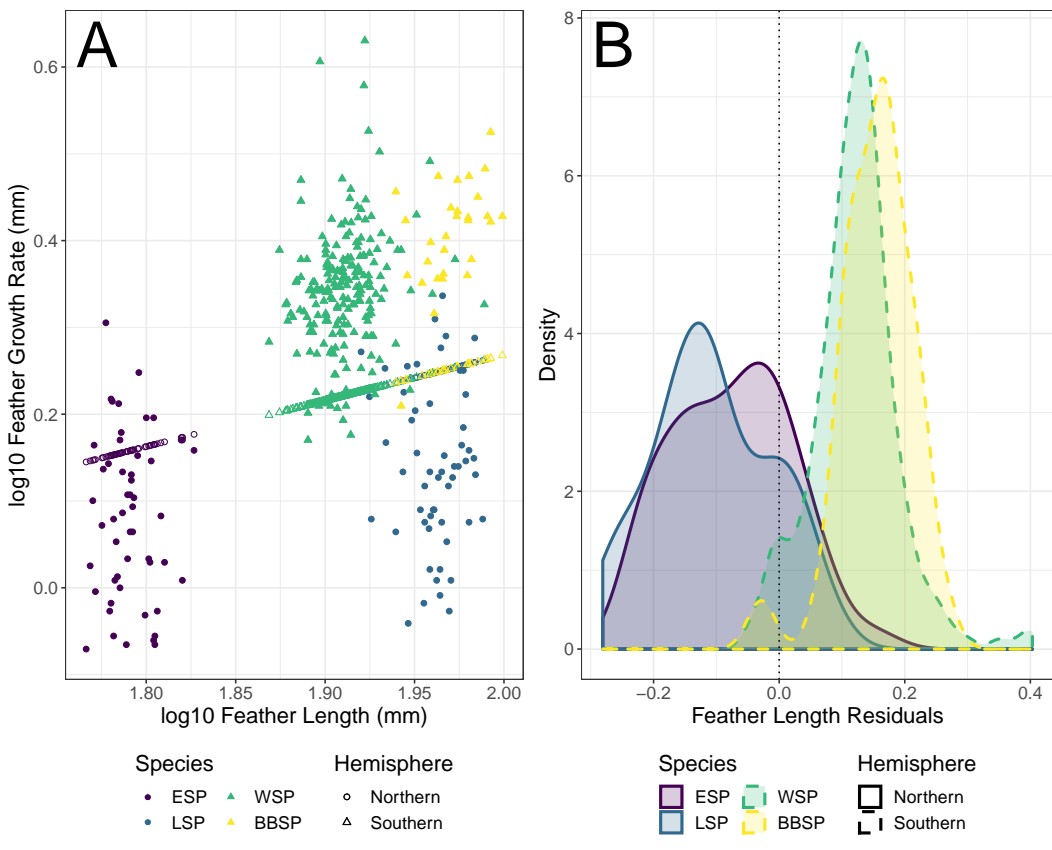

**Figure 3** **Predicted feather growth rates (FGR) and their residuals based on the optimized phylogenetic generalized least squares (PGLS) model for individual storm-petrels.** (A) Individual FGR was predicted (open points) using the optimized PGLS model (i.e., log10 FGR log10 FL + group, where FL is feather length and groups were specified as non-Procellariiformes vs. Procellariiformes), and residuals were calculated based on the distance to the observed values (closed points). Individual FL was used for model prediction and group was set to Procellariiformes. The Northern species are represented by dots, the Southern by triangles. European storm-petrels (ESP) are shown in purple; Leach's storm-petrels (LSP) in blue; Wilson's storm-petrels (WSP) in green; black-bellied storm-petrels (BBSP) in yellow. (B) The density plot shows residual distribution from the optimized PGLS model. The dotted line shows a residual length of 0.0. The Northern species are represented by solid lines, the Southern species by dashed lines. The colour codes are the same as in panel A. For statistical comparisons between the species and hemispheres see Table 5 & main text, and for species mean deviation from zero see Table 6.

In contrast, both Northern storm-petrels have been reported to show moult-breeding overlap, including the moult of tail feathers (*Ainley, Lewis & Morrell, 1976*; *Amengual et al., 1999*; *Arroyo et al., 2004*; *Bolton & Thomas, 2001*), though Leach's storm-petrels seem to start moulting relatively earlier in the breeding season than European storm-petrels. In the North Atlantic, around the Faroe Islands, primary production peaks over a longer period (*Eliasen, 2017*) as it is not linked to sea ice cover. Thus, food abundance might still be sufficient for moulting at the end of the breeding season for the Northern species. However, primary production varies strongly between years, which could lead to distinct inter-annual differences in food abundance for the storm-petrels (*Bonitz et al., 2018*; *ICES,*

*2005*; *ICES, 2008*). As food availability may thus be unpredictable for the Northern species during breeding, individuals may make different choices in prioritising either moult or reproduction, leading to obscured relationships between mean growth bar width and feather length.

*Langston & Rohwer (1996)* suggested that Laysan albatrosses (*Phoebastria immutabilis*) may form two growth bars per 24 h because their main prey (i.e., various squid species) is active at night and the albatrosses forage for them at dawn and dusk. This would result in two activity-rest cycles per 24 h, which would explain the formation of two growth bars daily as growth bar formation has been linked to sleep or rest rhythms (*Jovani et al., 2011*). Indeed, after doubling the feather growth rate of the Procellariiformes, their correlation between feather growth rate and feather length was very similar to that of the other orders (Fig. 2B), as shown by the lack of a significant group effect in the PGLSadj model. This seems to confirm *Langston & Rohwer*'s (*1996*) suggestion that Procellariiformes form two growth bars per 24 h. The four studied storm-petrel species may also have two activity-rest cycles per 24 h, which is consistent with the main prey activity of the studied storm-petrels during the breeding season, myctophid fish and krill. These prey species have a nocturnal activity similar to the prey of the albatrosses (*Hedd & Montevecchi, 2006*; *Siegel, 2012*), and several seabirds, including storm-petrels, forage in more oceanic habitats during the non-breeding period where they seem to increase their intake of myctophid fish (*Watanuki & Thiebot, 2018*).

We are aware of several possible limitations of the present study. Although growth bar widths have originally been linked to relative nutritional condition (*Grubb, 1989*; *Hill & Montgomerie, 1994*), it is not a direct measurement of food availability and the results should be interpreted with caution in that regard (*Murphy & King, 1991*). In this study we used feather growth rate as a way to retrospectively infer energy availability during moulting, as direct examinations of diet and food availability during moulting were impossible due to the pelagic nature and small body size of our study species. In order to put the feather growth rates observed in our study species into perspective, we compared their growth rates with data found in literature. However, while the reported measuring methods where similar between the studies, sampling techniques may have differed. In some studies samples were taken from museum specimens (e.g., *De la Hera, DeSante & Milá, 2012*; *Rohwer et al., 2009*) while others collected samples from live birds (e.g., *De la Hera et al., 2011*). This could lead to a bias in the condition of the birds sampled, as museum specimens may come from individuals in relatively poor health, or from relatively young individuals. Sample sizes per species ranged between 1 and 54 (*De la Hera, DeSante & Milá, 2012*) and for some species multiple sources were found (e.g., *Sitta carolinensis*: *De la Hera, DeSante & Milá, 2012*; *Dolby & Grubb, 1998*; *Grubb & Cimprich, 1990*. Especially Passeriformes where highly represented ($n = 160$ observations) in the model, while orders with larger species were under-represented. The model may therefore be less appropriate for larger species, but since the four storm-petrel species studied fall in a highly represented body size category in the model we feel it is appropriate to use here. Due to differences in the studied species' availability, large differences in sample sizes in our research occurred: the Northern species were comparatively abundant during mist-netting sessions, while

the Southern species were not. Additionally, nests of Wilson's storm-petrels were more accessible and concentrated than black-bellied storm-petrel nests, which were spread out over larger areas and more often located on inaccessible cliffs and ledges. Nevertheless, our study provides the first comparison of relative energy availability during tail-feather moult of storm-petrels differing in moult-breeding strategies and breeding in different hemispheres.

Our results suggest that for many pelagic seabirds ptilochronology may be a useful, non-invasive, and often only feasible, tool to study their relative energy allocation to feather growth during the non-breeding period when they are hardly accessible to researchers. Due to their specific life-history traits, pelagic seabirds may be especially interesting for ptilochronology studies as one may expect different patterns of feather growth compared to other species.

## CONCLUSIONS

We expected to find positive correlations between feather length and feather growth rate both within and between species, and to find differences in relative energy availability during moulting between species with differing moult-breeding strategies. The results of our analyses showed distinct differences in relative energy availability between four species of storm-petrels. The Southern species had a higher feather growth rate than predicted by a model based on data from multiple species and orders, while the Northern species had a lower feather growth rate than predicted. We suggest that all these differences can be attributed to the different moult-breeding strategies the species adopt, as the Southern storm-petrel species show no moult-breeding overlap while the Northern species do overlap both stages of the annual cycle. The better relative energy availability of the Southern species during moult may be explained by the fact that they change their feathers during the non-breeding period and can thus use all free energy for feather synthesis. In contrast, the Northern species have to allocate their energy between breeding and moulting. Our study shows how different moult-breeding strategies may affect the relative energy availability or energy allocation during moult of migratory pelagic seabirds. Additionally, we showed that at least a subset of the Procellariiformes likely forms two growth bars per 24 h instead of one, probably associated with the diel migration of their main prey species.

## ACKNOWLEDGEMENTS

We would like to thank the Henryk Arctowski Polish Antarctic Station and the Department of Antarctic Biology of the Polish Academy of Sciences for their hospitality and logistic support during our stay on King George Island. We are indebted to Jens-Kjeld Jensen and Marita Gulklett for their hospitality and help on the Faroe Islands. We are grateful also to the three anonymous reviewers of our paper that helped us improve its quality. Lastly, we would like to thank Rosanne Michielsen, Rachel Shepherd and Jessica Hey for their support in the field.

### Funding

This work was supported by a grant from the National Science Centre, Poland (2015/19/B/NZ8/01981). The funders had no role in study design, data collection and analysis, decision to publish, or preparation of the manuscript.

### Grant Disclosures

The following grant information was disclosed by the authors:
National Science Centre, Poland: 2015/19/B/NZ8/01981.

### Competing Interests

The authors declare there are no competing interests.

### Author Contributions

- Anne N.M.A. Ausems and Katarzyna Wojczulanis-Jakubas analyzed the data, contributed reagents/materials/analysis tools, prepared figures and/or tables, authored or reviewed drafts of the paper, approved the final draft.
- Katarzyna Wojczulanis-Jakubas analyzed the data, contributed reagents/materials/analysis tools, authored or reviewed drafts of the paper, approved the final draft.
- Dariusz Jakubas analyzed the data, contributed reagents/materials/analysis tools, authored or reviewed drafts of the paper, approved the final draft.

### Animal Ethics

The following information was supplied relating to ethical approvals (i.e., approving body and any reference numbers):

The Polish National SCAR, Institute of Biochemistry and Biophysics approved any harmful interference of Antarctic fauna (permit for taking or harmful interference of Antarctic flora and fauna: No. 7/2016 & No. 6/2017). On the Faroe Islands, birds were handled under license of the Statens Naturhistoriske Museum, Københavns Universitet (C 1012).

### Field Study Permissions

The following information was supplied relating to field study approvals (i.e., approving body and any reference numbers):

The Antarctic part of the study was performed under the permission of the Polish National SCAR, Institute of Biochemistry and Biophysics (permit for entering the Antarctic Specially Protected Area No. 3/2016 & No. 08/2017). On the Faroe Islands permission to enter the study site was sought through local land owners.

### Data Availability

The raw data are available in the Supplemental File.

## Supplemental Information

Supplemental information for this article can be found online at http://dx.doi.org/10.7717/peerj.7807#supplemental-information.

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
