# Peer review of "Differences in tail feather growth rate in storm-petrels breeding in the Northern and Southern hemisphere: a ptilochronological approach"

_PeerJ, doi:10.7717/peerj.7807_

## Round 0.1 · original submission · Major Revisions

General:

The paper is mostly well-written, but I think the take-home message is lost in the details. It seems there are competing models with different interpretations that should be discussed. Perhaps part of my confusion comes from your inability to directly compare the Procellariiformes-adjusted and outlier-adjusted models (see below for a suggestion).

Why use abbreviations for the species names? You’re not word limited. Spelling out the common names would improve the readability, especially for readers who are not familiar with storm-petrels.

Statistics:
Can you compare the three models using an information criterion such as AIC? That would allow a direct comparison between the Procellariiforms-adjusted and outlier-adjusted models.
Do the 2 adjusted models have different predictions? If so, please state and discuss them.

I’m not a fan of adjusting p-values for multiple testing, but there is not universal agreement on it. It looks like you have only one case where the adjustment matters (mGBW vs FM for Wilson’s storm-petrel). Please add a sentence to the discussion about this.

Editorial comments:
Line 77: Please insert “in” (“flexibility in allocation”).
Line 89: Insert “they” (“when they come”).
Line 115: Insert “of” (“each of the”).
Line 117: Insert “the” (“in the literature”).
Line 171: Insert “a” (“as a proxy”).
Line 196: Insert “were” (“values were averaged”).
Lines 292-293: I do not understand how this causes the formation of two growth bars per 24 h. Please clarify.
Line 305: Change “time-constraint” to “time-constrained”.
Line 308: Change “population” to “populations”.
Line 315: Insert “a” (“moulting a smaller number”).
Line 317: Change “time-constraint” to “time-constrained”.
Line 321: “Dissipate” is not the best word here. I suggest “weaken”.
Line 322: Change “constraint” to “constrained”.
Line 350: Change “restraint” to “constraint”.
Lines 350-352: This sentence is confusing. Please re-word.
Line 365: Change “from” to “form”.
Line 426: I think you mean “distances”, not “differences”.
Line 427: Change “constraint” to “constrained”.
Fig. 2 caption: “fitted values” implies the output of a model. Perhaps use “non-outlier”.

Figures:
Fig. 3: Can you make the histograms partially transparent, or plot the lines (but not the fill) on top of all the fill, so each entire histogram can be seen)?

Supplemental files:
Supplemental Table S1: Please add a caption to explain the headers (FGR, FL, FT).

Supplemental Data: Add a caption to explain the headers (FM, FL, mGBW) and explain the abbreviations under Species.

Reviewer 1 ·

Basic reporting

Generally well-written and clear. Adequate introductory and background material. Figures relevant and of good quality. Raw data supplied. Experiments appear ethical and supported by the appropriate permits.

Experimental design

Seemingly within the scope of the journal, although the importance of the knowledge gap being filled is somewhat unclear. Methods are described in sufficient detail.

Validity of the findings

Raw data appear to be sound but I have some serious concerns regarding the statistical approaches used (see general comments).

Additional comments

This paper addresses the relationships between various feather growth parameters in 4 species of storm-petrel. Variation in these species is then compared to data from a wider sample of 164 bird species, and the results of this broader comparative analysis are then used to inform further assessment of variation within storm-petrels.

Overall this study is well written and I liked the attempt to link patterns of feather growth rate within storm-petrel species to variation across birds more generally. However, while I have less of an issue with the within-species component, I have serious concerns with aspects of the comparative analyses that must be addressed before going forward.

First, I have a number of issues with the methods used to compare Procellariform growth patterns to those of other avian species. Most importantly, I think the approach of ‘adjusting’ data points and then comparing the fit of resultant models is inappropriate and statistically flawed. In particular, to my knowledge it is entirely inappropriate to use a likelihood ratio test to compare goodness-of-fit for two models that are based on different data (L215-218). In this case, one model is based on untransformed data and the second is based on a dataset where Procellariformes or all outlier FGRs have been adjusted by a factor of 2. These two models are based on different data sets and therefore statements such as “It [the model] was significantly improved by both Procellariiformes adjustment…” (L246-248) and “Indeed, the improved fit of the Procellariiformes-adjusted model compared to the original model seems to confirm Langston and Rohwer’s (1996) suggestion” (L369-371) are invalid because you are comparing two different datasets.

Rather than a priori ‘adjusting’ data points to what you believe they should be based on a hypothesis, I think a much better way to approach this analysis would be to let the data speak for itself. Specifically, I think it would be much better to compare the fit of a model in which all species are described by a single intercept and slope, to one in which the Procellariform species are allowed to have their own intercept. In modelling terms, the first model would be FGR ~ FL (i.e. model A in Fig. 2) and the second model in which FGR ~ FL + Group, where Group is a variable distinguishing Procellariforms from non-Procellariforms (i.e. everything else). It would then be valid to compare these models using likelihood ratio tests to see if the ‘Group’ term significantly improved model fit (although achieving statistical significance may be difficult due to low sample size [n=4] for the Procellariform group).

Nevertheless, if your hunch about Procellariform growth bars is correct, then the ‘Group’ term in the model should indicate a different intercept for Procellariform species versus everything else. Moreover, using this approach it should then also be possible to derive estimates for the intercepts associated with these two groups, and if your hypothesis is correct then the Procellariform intercept should be half that of other birds (presumably because Procellariforms grow two bands per day rather than one). To my mind, this approach represents a more direct test of your hypotheses that is both statistically valid and avoids the need for forcing a pre-defined ‘adjustment’ onto some of your data points – the ‘adjustment’ is instead estimated as part of the model.

However, I should also note that if you use the same approach to test for different intercepts between outlier versus non-outlier data points (i.e. groups indicated in Fig. 2C) then you will automatically find that outliers have a significantly lower intercept, because that’s how the points were identified in the first place (i.e. they were chosen because they were lower than everything else). This raises the question of whether this ‘outlier’ analysis is useful or even warranted in the first place.

Second, and more minor but still important, in L193 the authors’ state that the performed a PGLS analysis using the corBrownian function. This approach assumes that your traits (or more accurately the residuals of the relationship between your traits) perfectly fit a Brownian motion model of trait evolution. This is rarely the case and therefore it is often better to run a PGLS analysis using Pagel’s lambda model, which is optimised to reflect the level of phylogenetic signal in the data. This can be easily done in the same framework as currently used by using the corPagel function instead of corBrownian.

Minor comments

L67-87 – This may be a naive question but isn’t part of the explanation that individuals use plumage signals to attract mates and maintain pair bonds during breeding, therefore should not moult such signals until after the breeding season? This explanation also implies a trade-off, but not in terms of energy allocation as is assumed throughout.

L69 – Should this read ‘temporally separated’ rather than ‘temporarily separated’?

L111-117 – I felt that there was perhaps a sentence or two missing here regarding why this is an important/useful thing to do. Can the authors expand upon the knowledge gap being filled, and why it is important?

L154-161 – does year (2017/18) explain any of the variation in measurements?

L182 – delete unnecessary ‘the’.
L219 – I don’t really follow what is being done in this analysis.

Reviewer 2 ·

Basic reporting

I only assessed other basic aspects of the manuscript (see general comments for authors)

Experimental design

I only assessed some basic aspects of the manuscript (see general comments for authors)

Validity of the findings

I only assessed some basic aspects of the manuscript (see general comments for authors)

Additional comments

In this manuscript Ausems and coauthors explore how growth rate of tail feathers (obtained from ptilochronology) varied between four storm petrel species differing in the degree of moult-breeding overlap, i.e. two Northern hemisphere species typically showing this moult-breeding overlap, and 2 Southern hemisphere species that do not show it. They also frame the values for these 4 storm petrel species within a larger database to infer “the nutritional condition of this species in relation to other species”. I think this approach could be useful to make some inferences on the way these storm petrels moult (feather growth rate can be a surrogate of moult speed, De la Hera et al. 2011) and whether one feather growth bar corresponds to a 24-h period. However, ptilochronology-based estimates of feather growth rate cannot be unequivocally considered estimates of nutritional condition (particularly when comparing between species), since growth bar width is also affected by other factors (e.g. birds grow their feathers relatively faster under temporal constraints; De la Hera et al. 2009, 2012). This raises the possibility that storm petrels differ in their feather growth rate by other innumerable reasons apart from the existence of a breeding-moult overlap.
This study assumes implicitly that sampled storm petrels from the Northern populations experienced a moult-breeding overlap during the growth of their tail feathers, but I have some doubts about this. Arroyo et al. (2004) show that there is overlap between primary moult and breeding, but the overlap with tail feather moult is very small in the Mediterranean populations of European Storm petrels, and is expected to be even smaller in Northern conespecific populations (such as in Faroe Islands, Arroyo et al. 2004). Additionally, the birds sampled in the Northern hemisphere by Ausem and coauthors were trapped using mistnets (not in the burrows), so they are likely to be sub-adult birds (floaters) that, in fact, could have never bred before. These two circumstances suggest that the tail feathers analysed by the authors for the Northern Hemisphere species were not necessarily produced during breeding which, in turn, suggests that the differences in feather growth rate between species could be caused by other factors apart from this breeding-moult overlap.
For previous reasons, I think the study cannot be so strongly framed in studying the effects of the moult-breeding overlap on feather growth rate and, consequently, should be completely reformulated. I also added below a few other important (non-exhaustive) comments that arouse from a first reading of the manuscript:
Line 105-110: this statement is not correct and I think authors misinterpreted the current evidence. The interspecies relationship between feather growth rate and feather length is perfectly linear, but this relationship is not isometric; it typically shows a negative allometry where birds with longer feathers grow those same feathers comparatively slower than birds with shorter feathers. There is also a linear relationship between feather mass and length that is isometric in this case (see Rohwer et al. 2009, De la Hera et al. 2012).
Line 154: Unlike small passerines, plucking full tail feathers from non-passerines (especially large ones) might damage the feather follicle, which might prevent the growth of a new tail feather in the future. Do you have information on whether storm petrels are able to regrow their tail feathers after a manual extraction? This would be relevant for future studies from an ethical and animal welfare point of view.
Line 168-170: Feathers tend to grow slower at the beginning and at the end of their development, so growth bars should be measured in homologous regions of the feather to avoid potential biases (Grubb, 2006).
Line 168: one spare “the”.
Line 172-173: Defining feather quality as a ratio between feather mass and length is not a good idea (see Peig and Green 2009, Oikos 118: 1883-1891). Why not to analyse and plot the relationship between feather mass and length, similarly to what was done in Figure 2 and 3 for feather growth rate and length.

Reviewer 3 ·

Basic reporting

The manuscript “Does nutritional condition during tail moult differ in storm-petrels adopting contrasting moult-breeding strategies? A ptilochronological approach” by Ausems and colleagues, aimed to compare Growth Bar Width as a proxy for Feather Growth Rate in 4 species of storm-petrels: 2 breeding in the Northern Hemisphere with a breeding/moult overlap, and 2 breeding in the Southern Hemisphere with no breeding/moult overlap.
In general, the manuscript could benefit from a thorough check of the English grammar, for examples at lines: 77 “providing some flexibility allocation”; line 89 “when come to land”; line 196 “the values averaged per species”; line 304 “growing feathers faster lowers feather quality”.

The introduction nicely brought the subject of this study. I would make a stronger separation between the different costs of moulting and the costs of breeding (Line 59). I would also suggest the authors to use the same level of details throughout. For example, a species name is given Line 61, but not line 63.
I think the authors would benefit from reading Huntington et al. 1996 Leach’s storm-petrel. In The Birds of North America, No 233. The Academy of Natural Sciences, Philadelphia for a detailed account on the species.

Line 136. “move towards the equator” is not totally correct, as 2 of those species (LHP and WSP) are known trans-equatorial migrants (Drucker 2013, Wilson’s storm-petrel. In Neotropical Birds Online, Cornell Lab of Ornithology, Ithaca, NY; Pollet et al. 2019, Migration routes and stopover areas of Leach’s storm-petrels. Marine Ornithology 47: 55-65).

Line 415. Why not clearly state that is was among 4 species (rather than the more elusive between several species)?

Figure 1: I would include the significant correlation line for the Southern species.
Figure 3: The legend font is too small compare to the rest of the figure.

Experimental design

The methods sections would benefit from a few changes. For example, the sampling collection should be chronological. As it stands right now, the authors collected the feathers before the captured the birds.

Feather measurements: this section would require a bit more details on the methods, rather than simply citing Grubb (1989). A brief description on how the measure was done would be best.

Samples were collected in 2017 and 2018, but no mention was made on how authors dealt with pseudo-replication. Clarification is required.
Line 172, it should be “the ratio between feather mass and feather length” and not “the ratio between feather length and feather mass”.
Lines 190-195. This very long sentence would benefit from being cut into two distinct sentences. Furthermore, a more detailed description on the methods used for the literature search would be helpful.

Line 190. To better investigate nutritional condition during moult, I would suggest the authors to perform some stable isotopes analysis on the feathers. This would give them another variable to correlate with Growth Bar Width.

Is the Saino et al. 2012 reference regarding the correlation between FGR between primary and rectrices for all group of birds, or for a specific species (Barn swallow)? Does it take into account that northern storm-petrels moult their rectrices during the breeding season, but moult their primaries during migration, while barn swallow moult their rectrices and primaries during their wintering period? What are the results of the model if the feather types are included as a variable?

The aim of this study was to compare FGR between storm-petrels with different breeding/moult strategies, so I am not sure this phylogenetic analysis between FGR and FL is relevant for this study. It would be perhaps more relevant to have a comparison in FGR between species moulting while breeding and those moulting outside of the breeding season.

Supplementary table S1. As this table pertained to phylogenetic data, it would be helpful to have the species in phylogenetic order rather than alphabetical.

Validity of the findings

no comment

Additional comments

Line 132. As you have European storm-petrels, Leach’s storm-petrels, and Wilson’s storm-petrels, I think it would be best to have black-bellied storm-petrels (compared to black-bellied, Fregatta tropica (hereafter BBSP), storm-petrels).

Line 289-290. The comment that storm-petrels feed their chick during the night is correct, but the authors failed to take into account that individuals do not come back every evening to feed their chicks. According to Huntington et al 1996, for LSP, on average, chicks are fed by 1 adult on 44% of the nights, by 2 adults on 21% of the nights, and chicks are not being fed on 35% of the nights. As such interval between feeding is 2.2 d for a 30 to 40-day old chick and 2.8 d for a 60 to 70-day old chick (when the moult is happening).

Line 324. Line 421. There is some confusion about nutritional condition and energy allocation. Stating that southern species have a better nutritional condition (or a higher food availability as in line 421) might not be true, but it is true that they can allocate more energy to feather growth (as breeding is over).

Line 380 and Line 404. There is some discrepancies in the statement presented by the authors. Line 380: “Growth bar widths have been linked to relative nutritional condition,…. And results should be interpreted with caution” and the more assertive tone Line 404: “Our results suggest that ptilochronology may be a powerful non-invasive tool to reconstruct their relative nutritional condition…”

---

## Round 0.2 · Minor Revisions

I have a few editorial suggestions, and comments on the statistics. I like Reviewer 2's suggested title.
Line 115: You are now only comparing feather growth rate with feather length, so ‘feather characteristics’ should be ‘feather length’.
Line 151: ‘storm-petrel’ should be plural.
Line 174: ‘storm-petrel’ should be plural.
Lines 184-185: This is confusing. It sounds like capturing birds on the nest could lead to uncertainty in the breeding stage. I suggest ‘All individuals of the Northern species and some individuals of the Southern species were captured in mist-nets, which could lead to uncertainty ….’
Line 208: Delete extra period.
Lines 237-238: Would adjusting degrees of freedom for pseudo-replication affect your conclusions?
Lines 253-255: You should pick one method to evaluate your models. Using three methods only leads to confusion and possible contradictions. See below.
Lines 284-285: You could explain this here as you did in your response letter.
Lines 285: Please change ‘diurnally migrating prey’ to ‘prey that show diel migration’. Diurnal refers to daylight hours only, diel to the 24-hour cycle.
Lines 304-306: See comment for lines 253-255. The ANOVA (Likelihood-ratio test?) and AIC suggest that group is important, but that models 1 and 2 are equally supported. However, using only t- and p-values, one would conclude that group is important, but not the interaction term. Using all three criteria, I’m not sure which model is best.
Line 309-311: See previous comment. In addition, you report the AIC here as -501.47, and in Table 3 as -501.46. There might be something funny with the rounding. You report log likelihoods in Table 4 as identical for models 3 & 4, so ΔAIC should be 2.0. In this case, with ΔAIC = 1.99 or 2.0, I would argue that the simplest model (model 3) is best based on AIC. It’s also best based on t- and p-values, but not based on the ANOVA. Again, there is unnecessary confusion here. Best to stick with one method to compare models.
Lines 443-444: Please change ‘diurnal migration’ to ‘diel migration’.
Table 4: ‘No 1 is the full model, 2 & 3 with interaction and group dropped respectively’ is not quite correct. It implies that only interaction was dropped from model 2 and only group was dropped from model 3.
Supplemental Table S3: AIC is not comparable between different data sets, so there is no point reporting it here. I don’t know about Pagel’s λ.
Supplemental Dataset S1: The caption mentions FL and FGR. The table shows FM and FGR.

Reviewer 1 ·

Basic reporting

no comment

Experimental design

no comment

Validity of the findings

no comment

Additional comments

I am pleased to say that I find this paper much improved. I think the authors have adequately addressed the issues I raised in my previous review and because of it the paper is now much stronger. I commend the authors for their efforts in addressing these issues. Aside from this I have no further comments.

Reviewer 2 ·

Basic reporting

No comments

Experimental design

No comments

Validity of the findings

No comments

Additional comments

I read the new version of the manuscript by Ausems et al. after I provided comments on an earlier version of it before. I think the manuscript has improved and some of my remarks have been satisfactorily clarified. I am still not fully convinced about the strong focus provided to moult-breeding overlap as the main explanation for the observed variation in feather growth rate between North and South hemisphere storm petrels (although the title was changed the main storyline of the manuscript is the same), but authors acknowledge now in the text some of my hesitations (e.g. potential bias of the capture methods, limited knowledge of moult-breeding overlap in storm petrels). My only relevant concern is that I find a bit confusing the use of the term “nutritional condition” for the ptilochronology measurements of feather growth rate. As I mentioned before, growth bars can also reflect differences in temporal pressures so I do not find fully appropriate to use feather growth rate as a surrogate of nutritional condition, especially when the feather mass of the same feathers (as indicator of feather complexity) are ruled out from the analyses. Thus, differences in feather growth rate between storm petrel species could be explained by variation in feather complexity if the latter is negatively associated with feather growth rate between groups (see for example De la Hera et al. 2012 cited in the manuscript). In other words, feather mass can be relevant to distinguish whether feather growth rate varies because of nutritional differences between species or due to temporal constraints. This aspect also leads me to think about a more simple, and less confusing, title: “Does tail feather growth rate differ between storm-petrels breeding on the Northern and Southern hemisphere? A ptilochronological approach”.

---

## Round 0.3 · Minor Revisions

You’re almost there! Just a few more minor changes. Please get this back to me as soon as possible. I leave for the field 23 September.

Colors in supplemental figure S1: The caption describes the colors as purple and yellow, but they look blue and red.

Sample sizes: 229 WSP, 28 recaptured, leaves 200 unique individuals, not 201?

Adjusting p-values: Sorry, I was not clear enough in my prior comments. I want to know if adjusting p-values (Holm’s adjustment for multiple testing, lines 217-227 & 269) changes your conclusions. It seems most of your p-values are either low enough or high enough that it doesn’t matter whether they are adjusted for multiple testing or not.

Lines 233-234: If you use the information-theoretic approach (AIC comparisons) for model selection, you should not also use p-values for model selection. That is, simply run the all the biologically reasonable models and compare AIC among them. You’ve done this, but perhaps for the wrong reason. Just take out the part about dropping predictors that weren’t significant, state which models you ran, and give the AIC values.

Lines 276-278: I suggest “except between” instead of “but not between”.

---

## Round 0.4 · accepted · Accept

Thanks for getting it back so quickly.